# Discontent, Populism, or the Revenge of the "Places That Don't Matter"? Analysis of the Rise of the Far-Right in Portugal

Pedro Chamusca 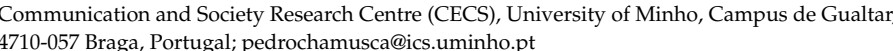

Communication and Society Research Centre (CECS), University of Minho, Campus de Gualtar, 4710-057 Braga, Portugal; pedrochamusca@ics.uminho.pt

**Abstract:** This research delves into the territorial nuances of political populism, examining Portugal's CHEGA party as a case study. Through a comprehensive analysis of survey data and correlational studies, this study reveals that discontent, manifesting in the rise of populist movements, is intricately linked to the economic decline and neglect of specific regions. The unexpected success of CHEGA is not merely a socio-economic phenomenon but a product of deeply rooted territorial dynamics. The findings underscore the importance of adopting place-sensitive development policies that address the unique challenges of overlooked territories, steering clear of traditional compensatory measures. The urgency to counteract long-term economic decline, industrial decay, and brain drain demands innovative strategies that tap into latent economic potential and provide tangible opportunities. As we confront the rise of anti-establishment voting threatening European unity, this research advocates for a paradigm shift towards place-sensitive policies to navigate the crossroads of discontent and foster a more resilient, inclusive future.

**Keywords:** political populism; territorial dynamics; place-sensitive development; geography of discontent

## 1. Introduction

Portugal celebrates fifty years of democracy this year (2024), restored in the country following the revolution on 25 April 1974, which ended the dictatorial regime known as the Estado Novo.

In terms of the country's governance, these fifty years have been marked by great stability, with the leadership of the Portuguese government alternating only between two political forces: the Socialist Party (a member of the Progressive Alliance of Socialists and Democrats) and the Social Democratic Party (a member of the European People's Party). The formation of the government, however, has been associated with some alliances (notably the Democratic Alliance, which supported the formation of right-wing governments) or parliamentary agreements (the most famous being the "geringonça", an agreement between the Socialist Party, the Portuguese Communist Party, the Left Bloc, and the Greens, allowing for the formation of a government by the socialists in 2015, despite losing the legislative elections that year). This stability is associated, among other aspects, with the importance of metropolization and suburbanization processes, combined with a strong dependence on community funds and periods of reduced social contestation, despite the continuous worsening of internal territorial asymmetries.

This governance stability has given way to a certain governmental monotony[1] [1,2], as since October 1995, the right has governed for just over seven years. In the 21st century, the Social Democratic Party (PSD) formed a government only four times: in 2002 with Durão Barroso (in early elections, following the resignation of António Guterres after a significant defeat in the 2001 local elections); in 2004 with Pedro Santana Lopes (replacing Durão Barroso, who left to preside over the European Commission), who remained in office for only eight months; in 2011 with Pedro Passos Coelho, who won the elections and

formed a coalition government after the fall of the socialist government led by José Sócrates following the request for external financial support; and in 2015 again with Pedro Passos Coelho, but he remained in office for less than a month as the formation of the government was not approved by the parliamentary majority formed by left-wing parties.

In this process of democratic consolidation, Portugal's accession to the European Union in 1986 proved decisive for the country's infrastructure, modernization, and administrative reform [3,4]. In fact, indicators of lag in various dimensions of development and the classification of practically all regions of the country as eligible for convergence/less developed[2] or ultra-peripheral[3] regions (with the exception of the Lisbon Metropolitan Area and the Algarve) justified a significant allocation of EU funds under different community frameworks and programs or policy instruments, with emphasis on the EU cohesion policy.

However, despite the integration of reformist principles in governance (participation, co-responsibility of agents, and governance model), the truth is that despite convergence on a European scale, development asymmetries between the various Portuguese regions continue to increase [2]. This trend of increasing regional asymmetries is often associated with the centralism of the governance system [1] and a distrust of governance institutions [5]. The first point concerns the concentration of much of the decision-making power being given to the central government with the absence of intermediate structures (regional or sub-regional), leaving municipalities more vulnerable and often without the competencies or financial resources for more efficient governance. In fact, several OECD documents reinforce this thesis, classifying Portugal as one of the most centralized countries. The second assertion is linked to a series of scandals involving high figures in governance, with a former Prime Minister (José Sócrates) being tried on suspicion of corruption, or the recent fall of Prime Minister António Costa, who resigned after learning that he was being investigated for suspected crimes of malfeasance, active and passive corruption of a political office holder, and influence trafficking.

Despite this scenario, Portuguese society remains relatively inexpressive in the organization of formal or informal movements of public dissent. The exception has been the struggles of professional organizations (teachers, nurses, and security forces, for example) or housing matters. In Portugal, the relatively low frequency of public protests can be attributed to several factors. Historically, the country experienced a long period of authoritarian rule under the Estado Novo regime, which suppressed dissent and left a legacy of cautiousness towards public demonstrations. Additionally, the strong influence of the Roman Catholic Church and a culture that values social harmony over conflict can contribute to a preference for resolving issues through dialogue rather than protest. Economic factors also play a role, as the focus on stability and recovery after the financial crises has led many to prioritize economic security over public activism. However, exceptions do exist, particularly among professional groups [6,7].

However, despite the apparent social calm, the far right has been progressively growing, becoming the third political force in the Portuguese parliament, increasing from one deputy in 2019 to twelve in 2022 and fifty in 2024. With the National Renewal Party being considered traditionally inexpressive, we have witnessed in recent years the emergence of CHEGA[4] (formerly known as BASTA), led by André Ventura, having an inflammatory, demagogic, and populist discourse marked by slogans associated with the fight against corruption and immigration. CHEGA, a far-right party in Portugal, shares similarities with previous far-right parties in its emphasis on nationalist rhetoric, anti-immigration policies, and criticism of mainstream political elites [8]. Like other far-right movements, CHEGA has capitalized on populist sentiments, portraying itself as a voice for the "ordinary people" against perceived threats to national identity and sovereignty [9]. While CHEGA's specific platform and tactics may differ from previous far-right parties, its emergence reflects broader trends of populist and nationalist sentiment gaining traction across Europe.

This article seeks to analyze and discuss the growth of the far right in Portugal by addressing two research questions: (i) What are the reasons explaining the growth of votes for the CHEGA party? (ii) What are the economic and social variables explaining the

territorially differentiated growth of this ideology? Through this analysis, we aim to discuss whether there is a geography of discontent [10–12], a revenge of the "places that don't matter" [13], or whether we are simply facing a process of media and social valorization of the populism and demagogy that characterize this discourse. This article is structured into six parts. After the Introduction, we present a literature review focused on the growth of far-right ideologies and Portuguese administrative reform. In Section 3, we present the research methodology, and in Section 4, we present the main research results. Section 5 presents a discussion, and the last section presents the conclusions.

## 2. Discontent, the Rise of the Nationalism, and the Specificities of Portugal

### 2.1. Geography of Discontent

The geography of discontent serves as a conceptual framework for comprehending the spatial dimension of societal dissatisfaction and political unrest within a given region or nation. This theoretical construct delves into the intricate interplay of geographical factors that contribute to the emergence and propagation of discontent among populations [14,15]. Rooted in political geography and sociology, the geography of discontent explores how spatial disparities, economic conditions, and regional inequalities intertwine with socio-political dynamics to shape the sentiments of the populace. Beyond a mere physical delineation of territories, this theoretical perspective invites an exploration of the socio-economic and political landscapes, unraveling the spatial intricacies that underpin the discontentment prevalent in societies.

At its core, the geography of discontent recognizes that discontentment is not a homogeneous force but rather a nuanced phenomenon influenced by geographic peculiarities. Factors such as economic downturns, regional imbalances, and perceived inequalities can manifest differently across diverse geographical locations, giving rise to unique expressions of dissatisfaction. This theoretical lens facilitates a comprehensive analysis of how localized grievances intertwine with broader socio-political narratives, leading to the emergence of movements, ideologies, or shifts in political landscapes [11,16,17].

Thus, discontent is a potent force that can shape a nation's political destiny. Discontent is becoming more and more relevant in Portugal due to recent social and economic issues and some generational changes. In fact, economic issues such as austerity measures, rising inequality, and housing shortages have led to increased public discontent. These pressing issues have mobilized people to take to the streets and demand change, highlighting the growing role of public dissent in shaping political outcomes. Younger generations, who did not experience the dictatorship, are more inclined to express their dissatisfaction and demand political changes. They are also more connected through social media, which helps them organize and amplify protests.

In the geography of discontent in Portugal, several pivotal factors come into play. A central point is the economic crisis that has engulfed the country over the past decade. The economic recession and austerity measures imposed have in response created a fertile ground for popular dissatisfaction. The perception of inequality and sense that political and economic elites do not represent the interests of the common populace have fueled the growth of discontent.

Another element to consider in the geography of discontent is regional division. In areas more profoundly affected by the crisis, such as the country's interior, unemployment and a lack of opportunities have risen, nurturing resentment against the status quo. Disparities in development between coastal and inland regions have been exploited by political movements as a means of mobilizing regional discontent in support of their agendas.

Additionally, the perception that traditional institutions are disconnected from the needs and aspirations of the population plays a significant role. The erosion of trust in democratic and party institutions has paved the way for populist and anti-establishment movements, positioning themselves as alternatives to conventional politics.

In short, the conventional characteristics of populist supporters relate to older age, lower education levels, and economic disadvantage [10,18–20]. This creates a "holy trinity"

of populist voting, traditionally associated with older, less-educated, and economically disadvantaged individuals [21–23].

*2.2. Rise of Extremist and Nationalist Movements*

Far-right parties typically advocate for conservative or reactionary positions, emphasizing nationalism, anti-immigration policies, and traditional values while often displaying authoritarian tendencies and prioritizing national sovereignty. On the other hand, populist parties appeal to the frustrations of ordinary people against elites, using anti-establishment rhetoric and claiming to represent the "true" voice of the people. While some far-right parties may incorporate populist strategies to broaden their appeal, populism itself can manifest across the political spectrum and is not inherently tied to specific ideological positions like those of far-right parties [24,25]. In Portugal, far-right parties totally incorporate populist strategies and discourses.

Traditional parties and institutions play a significant role in shaping the political landscape and public opinion. When these entities fail to adequately address societal crises or effectively respond to the concerns of the population, it can create a vacuum that populist and far-right parties often exploit [26]. Insufficient responses from traditional parties and institutions may include perceived or actual neglect of issues such as economic inequality, immigration, cultural identity, and political corruption. When people feel marginalized or disenfranchised by the mainstream political establishment, they may turn to populist or far-right alternatives that promise to address their grievances and provide simplistic solutions [27]. Additionally, a lack of trust in established institutions, coupled with a sense of frustration with the status quo, can further fuel the appeal of populist and far-right movements. These movements often capitalize on this discontent by presenting themselves as outsiders who will challenge the existing power structures and restore the voice of the "forgotten" or "ignored" segments of society.

The rise of the far-right in Portugal is intricately linked to the growth of extremist and nationalist movements. In recent years, we have witnessed a transformation in the political landscape with the emergence of parties and groups promoting radically nationalist, and at times, xenophobic agendas [28]. This phenomenon reflects a shift away from traditional political paradigms, driven by a confluence of factors that have stirred discontent among diverse populations.

The growth of extremist and radical political parties is associated with a variety of complex and interconnected factors. Although specific circumstances may vary depending on regional and national contexts, some common indicators and determinants include economic inequality, unemployment, and precarious work conditions. Economic crises and financial instability can also contribute to dissatisfaction, prompting people to seek radical political solutions [22–24]. Demographic changes, such as large-scale migrations, may create social and cultural tensions that extremists exploit.

Distrust in established political institutions can drive support towards radical alternatives promising substantial change. Issues surrounding cultural identity, ethnicity, or nationality may be leveraged by extremist parties to mobilize support [29]. Discontent with globalization and the perceived loss of national control can fuel support for radical parties advocating national interests. Insecurity and fears about security threats may lead to increased support for parties promising drastic measures to ensure safety.

One notable example is the rise of far-right movements in countries like Hungary, where Fidesz, led by Viktor Orbán, has adopted an openly nationalist and anti-immigrant stance. Orbán's government has implemented policies that focus on preserving what it perceives as Hungary's cultural and national identity, often employing rhetoric that frames immigration as a threat to these values. The party's dominance in Hungarian politics illustrates the success of such nationalist agendas in garnering public support.

In Italy, the League (Lega Nord), under the leadership of Matteo Salvini, has experienced a resurgence, aligning itself with populist and nationalist sentiments. Salvini's emphasis on immigration control and the prioritization of national interests has resonated

with a significant portion of the Italian electorate [17]. The League's success underscores how nationalist agendas can gain traction by capitalizing on economic insecurities and fears related to cultural identity.

The United Kingdom witnessed a seismic shift with the Brexit referendum, wherein the Leave campaign leveraged nationalist sentiments to advocate for the UK's departure from the European Union. While not a traditional political party, the Brexit movement tapped into a deep-seated dissatisfaction with perceived loss of control to Brussels, invoking notions of national sovereignty and reclaiming British identity. This example demonstrates how nationalist fervor can shape not only party politics but also influence major geopolitical decisions [10].

Moreover, the rise of far-right parties in France, exemplified by Marine Le Pen's National Rally (formerly National Front), illustrates the enduring appeal of nationalist narratives. Le Pen's campaigns have focused on anti-globalization sentiments, concerns about cultural preservation, and skepticism towards immigration, tapping into the discontent of those who feel left behind by globalization and European integration.

The rise of CHEGA in Portugal shares some similarities with the political landscape in these countries. It also connects strongly with some contemporary developments like in the Netherlands and Sweden, where far-right parties have also experienced varying degrees of growth. In the Netherlands, the Party for Freedom, led by Geert Wilders, has gained prominence with its anti-Islam and anti-immigration stance, appealing to voters disillusioned with mainstream politics. Similarly, in Sweden, the Sweden Democrats have capitalized on concerns about immigration and multiculturalism to become a significant political force [7]. These parties, like CHEGA, have tapped into populist sentiments and criticized traditional political elites for failing to address the anxieties of the population. While the specific contexts and issues may differ across these countries, the broader trend of far-right parties gaining ground by exploiting societal tensions and grievances against established institutions is evident [6].

These instances collectively illuminate a broader trend of nationalist resurgence, where political actors exploit discontent by framing it within a nationalist narrative. Economic uncertainties, perceived threats to cultural identity, and fears associated with immigration have become potent tools in the arsenal of these movements. While not uniform across Europe, this wave of radical nationalism poses challenges to the established political order, underscoring the need for a nuanced understanding of the geographical and societal factors that fuel discontent and shape the rise of such political forces.

Within this resurgence of nationalism, the refugee crisis and immigration have been central themes in the rhetoric of these movements. The exploitation of fears surrounding the loss of cultural identity and the supposed threat that immigration poses to local resources and jobs has become a recurring strategy. Nationalism is often used as an appeal to preserve cultural and ethnic "purity", resonating with a disenfranchised segment of the population. Furthermore, the spread of fake news and misinformation has played a crucial role in the expansion of these movements. Social media platforms have become fertile grounds for the propagation of extremist ideas, nurturing information bubbles that reinforce radical beliefs. Easy access to distorted information has contributed to the creation of a fervent and, at times, radical support base.

The crisis of trust in democratic institutions is also a central factor in the rise of these movements. The perception that representative democracy is tainted and ineffective propels the appeal for authoritarian leadership and radical solutions [1,30,31]. Disenchantment with conventional politics leads many to seek more extremist alternatives, presenting themselves as the voice of the people against an ostensibly corrupt elite.

### 2.3. (De)centralization and Governance Context in Portugal

Portugal is a country with a population of 10,343,066 inhabitants, located in Southwestern Europe. As a member of the European Union since 1986, Portugal has been implementing a series of administrative reforms over the past decades with the aim of

promoting principles such as the simplification of decision-making processes, greater openness and transparency in procedures, and increased involvement and co-responsibility of different stakeholders [5]. However, there is still a long way to go.

Given the importance of decentralization for development and wellbeing, a recent study by the OECD [32] identifies Portugal as one of the countries with a more centralized state (Figure 1), highlighting the advantage of territorial-based development to strengthen subsidiarity, autonomy, and efficiency in public policies. The successive international economic crises (strongly felt in Portugal), growing housing access problems, and the consolidation of demographic, social, and economic processes, such as depopulation, aging, uneven population distribution in the territory, and the loss of qualified population, only partly explain the problem. However, the fact that subnational public spending in Portugal is among the lowest in the OECD primarily results from the centralism that characterizes the Portuguese governance model, increasing the distance from elected officials and their distrust regarding a set of governance options.

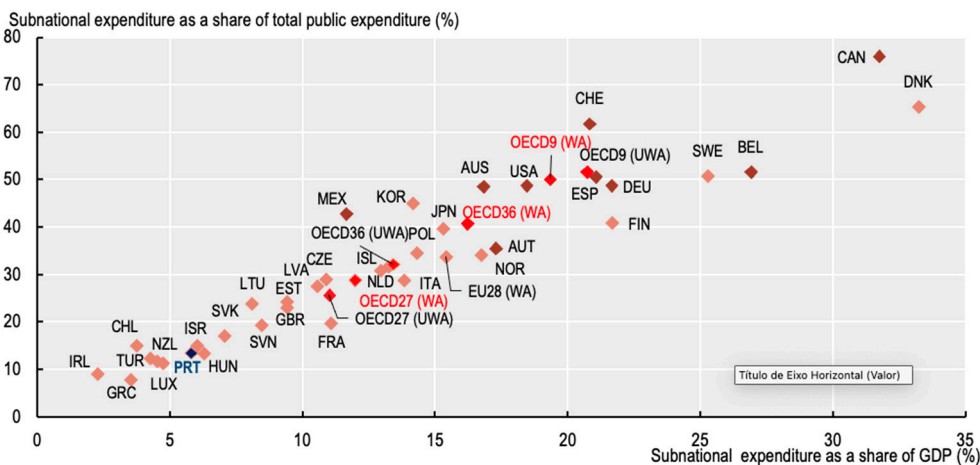

**Figure 1.** Subnational government expenditure as a percentage of GDP and total public expenditure, 2018. Source: [32].

This context helps us to understand that a recent study [33] has shown that over 60% of Portuguese citizens do not trust the Assembly of the Republic, a figure that is above the EU average. According to the report analyzing the perception of Portuguese people regarding politics conducted by the statistical database of the Francisco Manuel dos Santos Foundation, which uses data from the 2023 Eurobarometer, eight out of ten respondents in Portugal tend not to trust political parties. This aligns with the trend in 19 out of the 27 European Union (EU) countries, where over 70% of people tend not to trust political parties.

The data from the European Social Survey reveal that Portugal is among the four countries where citizens have the least confidence in their ability to participate in politics (83%), following Slovakia (84%) and alongside Latvia and the Czech Republic (83%). Regarding political participation, the study highlights that 73% of national citizens believe that the system does not allow or allows very few people to influence politics. This perception is shared by over half of the countries analyzed, except for Norway, Switzerland, Finland, Iceland, and the Netherlands, where it is considered that political systems allow people some degree of influence.

### 3. Methods

For this investigation, we employed two methods of analysis: a survey and a statistical analysis, correlating the voting patterns of CHEGA (and its growth) with economic and social variables.

The survey was conducted online using the Survey123 application. It was administered in the five municipalities where CHEGA obtained the highest percentage of votes in

the legislative elections of 2022, namely Serpa, Moura, Salvaterra de Magos, Vila Nova da Barquinha, and Vila Verde[5]. The survey took place between 1 September and 30 October 2023 and was conducted entirely anonymously, in accordance with the General Data Protection Regulation in force in Portugal. The first question recorded the voting preference in the legislative elections of 2022, considering only respondents who admitted to voting for the CHEGA party in this investigation. It intended to analyze the perceptions, evaluations, and voting intentions of the electorate in CHEGA. The questionnaire was made freely available, the response was not limited to a specific profile (demographic, social, or economic), and only responses from those who confirmed having voted for CHEGA in 2022 were validated. Based on this response, we obtained the sample as shown in Table 1.

**Table 1.** Survey sample (valid answers). Source: own elaboration.

| Municipality | Nr. of Answers |
| --- | --- |
| Serpa | 21 |
| Moura | 13 |
| Salvaterra de Magos | 43 |
| Vila Nova da Barquinha | 19 |
| Vila Verde | 21 |
| Total | 117 |

Thus, the applied survey recorded 117 valid responses from participants who acknowledged voting for CHEGA in the legislative elections of 2022. The profile of the respondents was as follows:

- Sex: male (63%), female (35%), other or prefer not to respond (2%);
- Age: under 30 years old (33%), 31 to 49 years old (26%), 50 to 64 years old (18%), or 65 or older (23%);
- Qualifications: higher education (31%), secondary education (42%), third cycle of basic education (17%), lower or no qualifications (9%), or did not respond (1%);
- Employment status: public sector employees (38%), private sector employees (23%), self-employed (19%), students (14%), unemployed (4%), or did not respond (2%).

The profile was recorded solely for sample categorization, not intending to guide the analysis of the response, as it is impossible to determine its representativeness of CHEGA voters. In this survey, responses to the following questions were considered, all of which were closed-ended, allowing the selection of one or more options:

1. Was it the first time you voted for CHEGA? In case of previous votes, in which elections did you do so?
2. What are the main reasons for voting for CHEGA?
3. How do you assess the performance of CHEGA in parliament?
4. Do you intend to vote for CHEGA in the upcoming elections? Why?
5. Indicate your level of agreement with CHEGA's proposals regarding: (i) combating corruption; (ii) reducing state welfare policies; (iii) reducing subsidies to institutions and individuals; (iv) immigration control; and (v) strengthening security.

The second analytical component sought to determine if there was any social or economic pattern that explained the growth of votes for the CHEGA party. A total of 19 variables (plus 3 associated with votes) were selected (Table 2), and a correlation analysis was performed with the percentage of votes obtained by CHEGA in the legislative elections of 2024 and with the growth in the percentage of votes relative to 2022 in each of the 308 municipalities in Portugal.

**Table 2.** Correlation variables. Source: own elaboration.

| |
| --- |
| Proportion of votes for the CHEGA party in the legislative elections of 2022/2024 (%) |
| Variation in CHEGA's election results (percentage of votes) between the legislative elections of 2022 and 2024 |
| Abstention rate in the legislative elections of 2022/2024 (%) |
| Inhabitants (2021, nr.) |
| Population density (2021, nr./km$^2$) |
| Settlement types (under/above 2000 inhabitants, 2021) |
| Population dimension (groups, 2021) |
| Proportion of resident population with foreign nationality (%) |
| Proportion of resident population with completed higher education (%) |
| Proportion of resident population with only the first cycle of basic education completed or without any level of education (%) |
| Proportion of resident population aged 20–29 years (%) |
| Proportion of resident population aged 65 years or more (%) |
| Average monthly income of the resident population (EUR) (2022) |
| Income inequality (2022) |
| Unemployment rate (%) |
| Recipients of social insertion income from social security per 1000 active-age inhabitants (‰) |
| Crime rate (‰) |
| Gini coefficient of gross income declared by fiscal household (%) |
| Purchasing power (2021) |
| Public investment (EUR/person) (2021) |
| Funding executed under Portugal 2020 (EUR) (2024) |
| Funding executed per capita under Portugal 2020 (EUR/person) (2024) |

The correlation sought to assess any statistical relationship (causal or non-causal) between two variables. The Pearson correlation coefficient was calculated according to the following formula:

$$\rho = \frac{\sum_{i=1}^{\eta}\left(x_{i-\bar{x}}\right)\left(y_{i-\bar{y}}\right)}{\sqrt{\sum_{i=1}^{\eta}\left(x_{i-\bar{x}}\right)^2}\cdot\sqrt{\sum_{i=1}^{\eta}\left(y_{i-\bar{y}}\right)^2}} = \frac{cov\ (X,Y)}{\sqrt{var(X).var(Y)}}$$

The Pearson correlation coefficient measures the linear relationship between two variables. To determine the statistical significance of this correlation, we need to test the hypothesis that there is no correlation between the variables (0). The Pearson correlation coefficient provides a measure of the strength and direction of the linear relationship between two variables. Values close to $\pm 1$ indicate strong relationships, while values close to 0 indicate weak or no linear relationship. The value of 1 indicates a perfect positive linear relationship between the two variables. As one variable increases, the other variable increases in exact proportion. The value of $-1$ indicates a perfect negative linear relationship. As one variable increases, the other variable decreases in exact proportion.

We conducted Grubbs's test to identify potential outliers in our dataset, which comprised information from 308 municipalities. The purpose of the test was to determine if any data points significantly deviated from the others, potentially affecting our analysis. The results of Grubbs's test indicated that the presence of outliers was minimal, suggesting a high level of consistency and reliability in our data.

- Test statistic (G-value): 1.85;
- Critical value: 2.50;
- *p*-value: 0.08;
- Decision: no significant outliers detected.

Consequently, the test confirmed that the influence of outliers on our overall findings was negligible, allowing us to proceed with confidence in the robustness of our dataset.

The results were also mapped using ArcMap 10.8.2 software from ESRI.

## 4. Results

### 4.1. Territorial Patterns

The rise of far-right political sentiment in Portugal, particularly within the context of legislative elections, has been accelerating at a remarkable pace. Traditionally marginal, with the National Renovator Party gradually gaining votes from 2002 to 2015 and peaking at 27,269 votes, which corresponded to 0.5% of the total vote, the emergence of CHEGA and its leader André Ventura has revolutionized this landscape. Contesting legislative elections for the first time in 2019, CHEGA secured a parliamentary seat (André Ventura) in the Lisbon electoral district, garnering 1.29% of the nationwide vote (67,826 votes). In 2022, its vote share surged to 7.18% (399,510 votes), establishing itself as the third-largest political force in parliament with twelve elected deputies. The 2024 elections witnessed the historic zenith of CHEGA's growth in Portugal, solidifying its position as the third major political force, closely trailing the ruling parties (29.5% and 28.7%), securing 18.06% of the votes (1,108,797 voters) and electing 50 out of 230 members to the Assembly of the Republic (Figure 2).

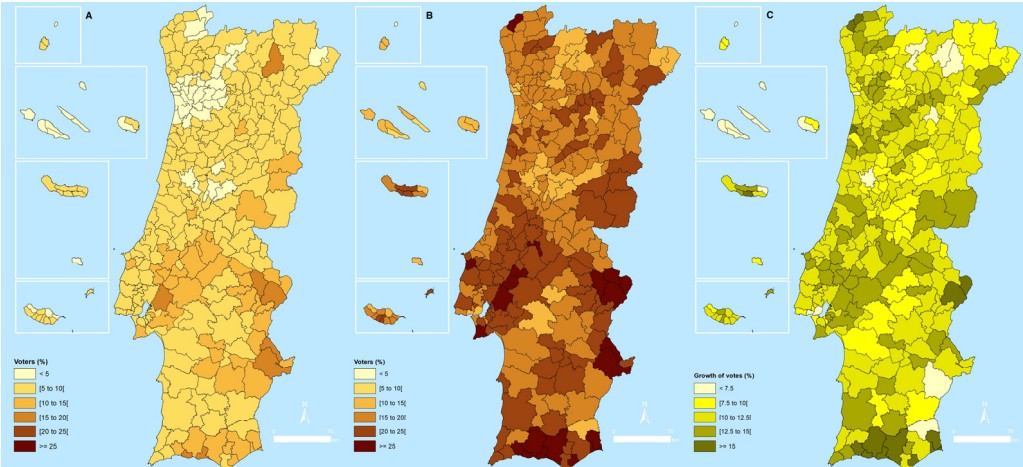

**Figure 2.** Votes in the CHEGA party: (**A**)—% of votes in 2022; (**B**)—% of votes in 2024; (**C**)—% of growth 2022–2024. Source: [34].

In an election with a 33.8% abstention rate (the lowest since the 1995 legislative elections) and a record 6.1 million voters, CHEGA's expansion was both substantial and widespread, failing to secure parliamentary representation only in the Bragança district. In fact, it garnered less than 15% of votes in only 32 municipalities, exceeded 30% in five, and dipped below 10% only in the city of Porto in mainland Portugal. The highest vote percentage was recorded in Elvas, Portalegre district, with 36.53% (4150 votes). Areas that followed closely were Albufeira, Lagoa, and Portimão in the Faro district, with 32.61% (6774 votes), 31.14% (4022 votes), and 30.53% (9705 votes), respectively. In the Alentejo region, it achieved 30.45% (2156 votes) of the vote in the Moura municipality (Beja). Noteworthy results were also observed in Benavente (Santarém), Silves (Faro), and Mourão (Évora), with 29.98% (4902 votes), 29.97% (5796 votes), and 29.77% (407 votes) of the votes, respectively. Valença (Viana do Castelo) ranked 9th with 29.56% (2215 votes), followed by Olhão (Faro) with 29.33%.

In the Faro district (Algarve), CHEGA emerged as the leading political force, electing three out of nine deputies. Post-election media analyses indicated that the growth of CHEGA was grounded in the perceived failure of the PS government to address various life expectations of the population, notably in healthcare (insufficient doctors in hospitals, fortnightly closures of pediatric and obstetric emergency services), education (shortage of teachers), and living conditions (lack of housing solutions, rising living costs associated with inflation, low wages, precariousness, and unemployment linked to the dominant tourism sector). The growth of CHEGA presents different territorial patterns (Figure 2), despite

the results of the latest legislation pointing to greater spatial coverage and affirmation throughout the national territory.

The rise in votes for CHEGA can be attributed to a combination of socio-economic factors, political disillusionment, and cultural anxieties, considering the main rationale of public discussion within the media and academic meetings. Firstly, economic uncertainty and inequality have left many Portuguese citizens feeling marginalized and overlooked by the mainstream political establishment. CHEGA's nationalist rhetoric and promises to prioritize the interests of ordinary Portuguese people resonate with those who perceive themselves as being left behind by globalization and economic reforms. Secondly, there is a growing disillusionment with traditional political parties, which are seen as out of touch with the concerns of the average citizen and mired in corruption scandals. CHEGA's leader, André Ventura, has capitalized on this sentiment by positioning himself as an outsider willing to challenge the status quo and shake up the political establishment. Additionally, cultural anxieties about immigration and national identity have fueled support for CHEGA, as the party advocates for stricter immigration policies and portrays itself as a defender of Portuguese heritage and values. By tapping into these societal tensions and offering simplistic solutions to complex issues, CHEGA has managed to attract a significant share of the electorate, particularly among those disenchanted with mainstream politics.

### 4.2. The Voters' Opinion

The electorate of the CHEGA party has been growing, which is associated with its ability to capture new voters. Of the 117 respondents, 68% had never voted for CHEGA before, doing so for the first time in the 2022 electoral act. Among those who had previously voted for CHEGA, the majority had expressed their vote for the party in previous legislative (84%) and/or European (61%) elections, with a smaller number of respondents also having voted for the party in local elections (Figure 3), which seems to indicate a lower capacity to project the party's ideas and figures in local electoral acts (where they do not preside over any municipal council).

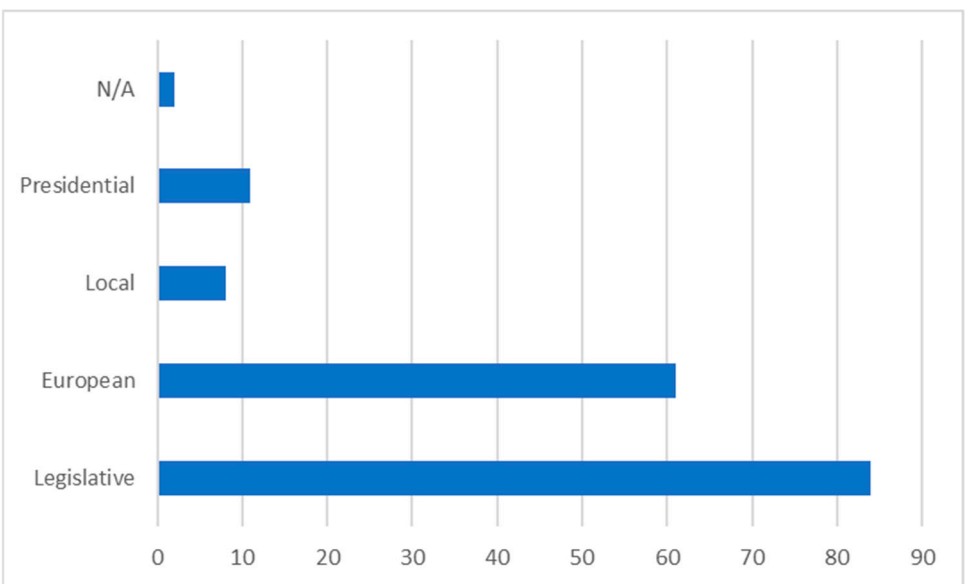

**Figure 3.** Votes in the CHEGA party before the 2022 legislative election. Source: own elaboration, using data from questionnaires.

The assessment made by the CHEGA electorate regarding the performance of the parliamentary group elected in 2019 and 2022 was overwhelmingly positive. In fact, 34% of respondents rated the performance as excellent and 57% as good, with 7% rating it as fair and only 1% as poor (Figure 4). It is therefore not surprising that the vast majority of the electorate remains loyal to the party, with 97% admitting that they will maintain their

vote in the next elections (organized on the 10 March 2024, following the dissolution of the Assembly of the Republic and the fall of the government, a fact not known at the time of the survey).

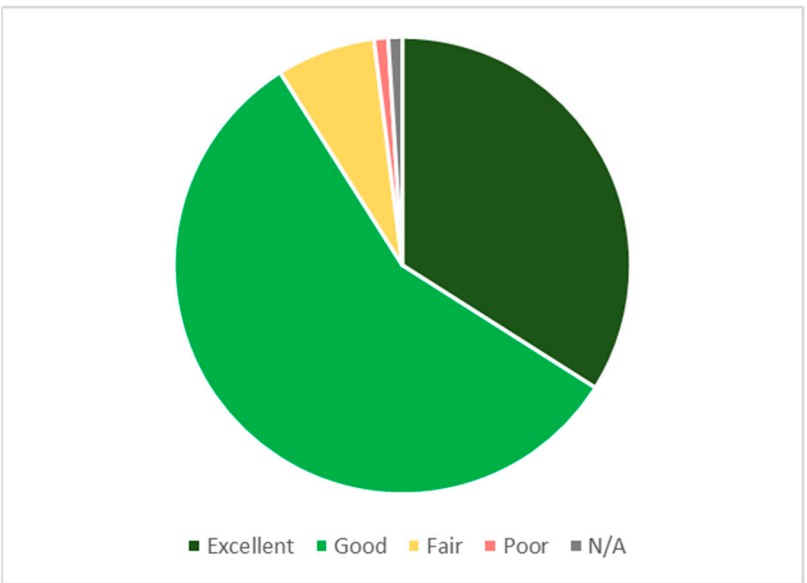

**Figure 4.** Evaluation of the CHEGA party parliamentary performance. Source: own elaboration, using data from questionnaires.

We sought to analyze the underlying reasons for this choice (Figure 5). The majority of responses point towards the appreciation of radical promises in the discourse of the far right while downplaying the integrated strategy component for the country. People are drawn to vote for the right-wing party CHEGA in Portugal, as well as similar parties across Europe, due to a combination of factors reflecting widespread discontent and a desire for change. The survey data indicate that issues such as perceived corruption within the government, disillusionment with traditional political parties, and concerns about economic and social degradation strongly influence voter choices. The political left has been unable to talk on these issues and provide a powerful alternative narrative, as they have supported the government between 2015 and 2019 and some of their policies after that. The figure of André Ventura, the party leader, plays a significant role, suggesting the appeal of charismatic leadership. Moreover, the emphasis on combating corruption, reducing dependency on subsidies, and controlling immigration aligns with a broader trend seen in right-wing movements across Europe.

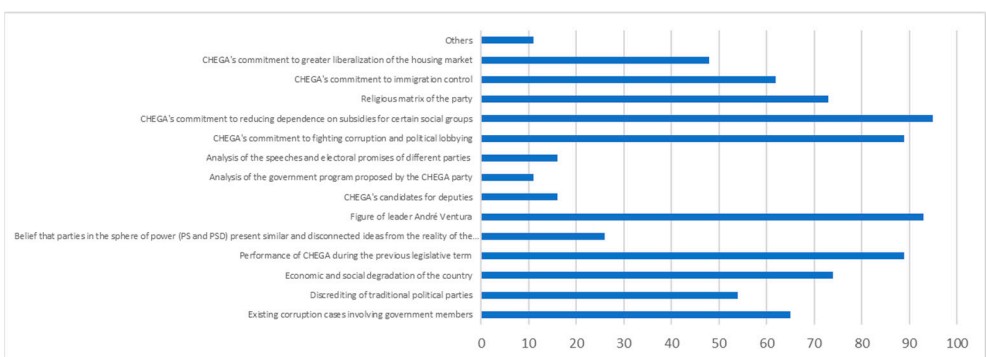

**Figure 5.** Reasons for voting in the CHEGA party. Source: own elaboration, using data from questionnaires.

The survey revealed that 65% of respondents voted for CHEGA due to existing corruption cases involving government members, reflecting a pervasive concern about ethical conduct within the political sphere. Furthermore, 54% expressed a general disillusionment with traditional political parties, highlighting a broader trend of distrust in established political entities. The economic and social degradation of the country emerged as a compelling factor for 74% of voters, emphasizing the urgency for change.

Notably, a striking 89% based their vote on the perceived performance of CHEGA during the previous legislature, showcasing the impact of the party's actions in shaping voter preferences. Conversely, only 11% considered the analysis of the proposed government program, indicating a potential reliance on party reputation rather than policy scrutiny.

Interestingly, the figure of leader André Ventura played a pivotal role, influencing a significant 93% of respondents. This strong leadership appeal may suggest a cult of personality, indicative of the party's charismatic leadership strategy. Moreover, the emphasis on combating corruption and reducing dependency on subsidies garnered substantial support, influencing 89% and 95% of respondents, respectively, showcasing a demand for accountability and fiscal responsibility among the electorate.

Thus, the survey illuminates a complex landscape of voter motivations for choosing CHEGA. While some prioritized the party's specific policy positions, others seem swayed by broader sentiments such as disillusionment with traditional politics, anti-corruption stances, and a charismatic leadership figure. The findings underscore the multifaceted nature of political choices and highlight the challenges in addressing voter concerns through comprehensive policy analysis and communication. This is also evident in the strong agreement with majority of the party's most relevant proposals (Figure 6), with particular emphasis on combating corruption and reducing the number of beneficiaries of public subsidies.

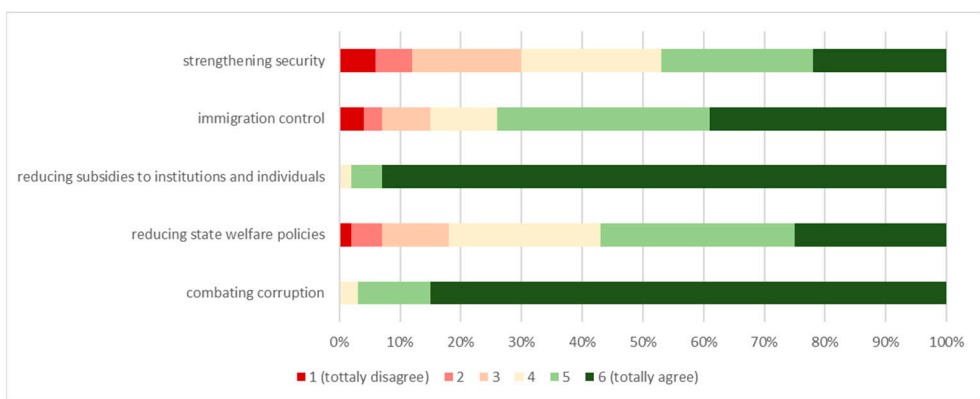

**Figure 6.** Agreement with the CHEGA party proposals. Source: own elaboration, using data from questionnaires.

### 4.3. The Drivers of Far-Right Growth

#### 4.3.1. General Analysis

The analysis conducted reveals a relatively low correlation between the voting for the CHEGA party and a set of demographics, social, economic, or decentralized or place-based public investment indicators (Table 3).

In fact, in the 2022 legislative election, a low correlation was observed for five indicators: number of beneficiaries of social insertion income, percentage of resident population of foreign nationality, unemployment, settlement types, and the public funding under the EU communitarian funds. A moderate correlation was observed with crime rate data (Figure 7). This suggests a greater implantation capacity of the CHEGA party in municipalities where some of its electoral priorities—the need to control immigration to reduce crime and the commitment to combating corruption with greater control over the allocation of subsidies and social support—end up having a more practical expression. Thus, the voting for the CHEGA party tends to grow proportionally to the growth of these mentioned indicators,

highlighting a direct relationship with social issues. The voting also tends to diminish as the public funding increases.

**Table 3.** Correlation analysis of votes (global) with demographics and economic and social data. Source: own elaboration, using data from [34,35] [*p*-value: 0.09].

|  | **Votes 2022** | **Votes 2024** | **Growth 2022–2024 (%)** |
|---|---|---|---|
| Abstention (2022) | −0.03 | −0.14 | −0.13 |
| Inhabitants (2021) | −0.12 | −0.13 | −0.10 |
| Density (2022) | −0.14 | −0.18 | −0.16 |
| Settlement types (2021) | 0.20 | 0.18 | 0.09 |
| Population size (2021) | −0.07 | 0.00 | 0.08 |
| Foreigners (2021) | 0.26 | 0.30 | 0.24 |
| % of 20–29-year-old inhabitants (2021) | −0.16 | −0.01 | 0.16 |
| % of 65+-year-old inhabitants (2021) | 0.05 | −0.06 | −0.17 |
| % of low-qualification inhabitants (2021) | −0.14 | −0.16 | −0.13 |
| % of higher education inhabitants (2021) | 0.01 | −0.07 | −0.15 |
| Social income (2022) | 0.27 | 0.19 | 0.04 |
| Crime rate (2022) | 0.41 | 0.38 | 0.21 |
| Average earnings (2021) | 0.01 | −0.02 | −0.06 |
| Income inequality (2021) | 0.08 | −0.01 | −0.11 |
| Unemployment (2021) | 0.24 | 0.30 | 0.28 |
| Gini coefficient (2021) | 0.01 | −0.04 | −0.10 |
| Purchasing power (2021) | 0.09 | 0.04 | −0.04 |
| Public investment (2021) | −0.02 | −0.15 | −0.27 |
| Funding executed under Portugal 2020 (2022) | −0.22 | −0.23 | −0.17 |
| Funding executed per capita under Portugal 2020 (2022) | −0.15 | −0.28 | −0.34 |
| | Very low | Low | Moderate |

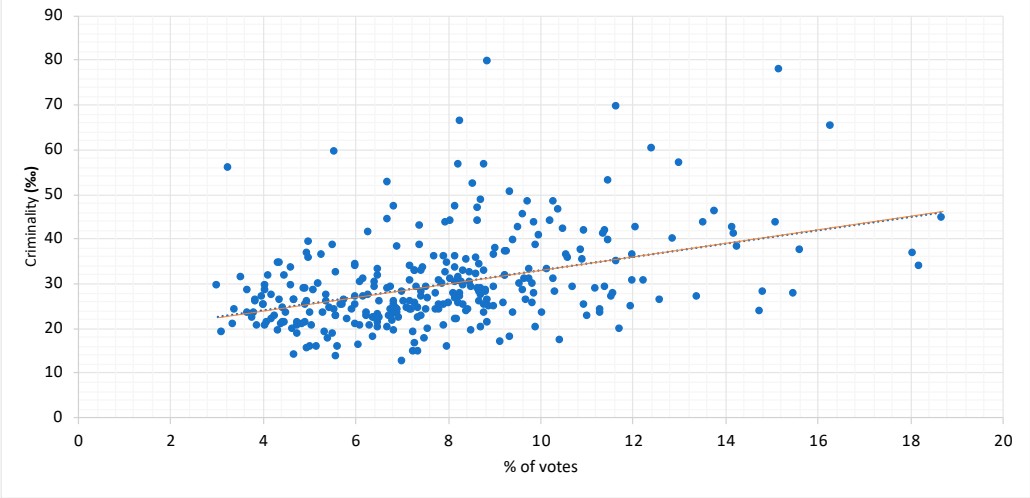

**Figure 7.** Moderate correlation of votes (global, 2022) with crime rate. Source: own elaboration, using data from [34,35].

In the electoral act of 10 March 2024, we observe some differences. Despite crime and the percentage of foreign population continuing to show a weak correlation, other factors gained prominence, notably two factors associated with the discontent of territories

that tend to be overlooked due to the concentration of public investment in major urban centers. Indeed, there was an increase in CHEGA party votes in municipalities with higher unemployment rates, while the vote decreased in councils that had a higher volume of per capita public investment executed locally under the Portugal 2020 framework. The issue of the foreign population remains relevant, despite the low correlation indicators. In the sixteen municipalities with a proportion of foreign population exceeding 10%, CHEGA obtained an average vote of 23.6% (higher than the 18.09% overall vote), with four of them ranking among the seven highest percentage votes for the party nationwide.

When analyzing the relationship between the growth rate of CHEGA party votes between 2022 and 2024 (legislative elections) and these indicators, we find that social issues tend to lose some relevance, giving way to a certain revolt in neglected territories. The identified moderate correlations are directly associated with the outcome of certain public investment choices, with growth being higher in territories with higher unemployment and in municipalities with lower public investment or approved investment volume under the Portugal 2020 community framework. These facts demonstrate that, despite the widespread growth in party voting, discontent takes on different geographical expressions, associated with a certain revolt in overlooked territories [13].

### 4.3.2. Cluster Analysis

The analysis of voting across the 308 municipalities highlights some weak and moderate correlations. When we narrowed down the scale of analysis to the thirty municipalities with the highest/lowest vote share in 2002 and 2024 or to the variation between 2022 and 2024, we were able to extract some additional information (Table 4).

**Table 4.** Correlation analysis of votes (cluster) with demographics and economic and social data. Source: own elaboration, using data from [34,35] [*p*-value: 0.07].

| | 2022—30 Municipalities with Higher % of Votes | 2022—30 Municipalities with Lower % of Votes | 2024—30 Municipalities with Higher % of Votes | 2024—30 Municipalities with Lower % of Votes | 2022–2024—30 Municipalities with Higher % of Growth | 2022–2024—30 Municipalities with Lower % of Growth |
|---|---|---|---|---|---|---|
| Abstention (2022) | 0.26 | −0.08 | 0.41 | −0.03 | 0.29 | 0.18 |
| Inhabitants (2021) | −0.04 | 0.15 | 0.19 | −0.03 | −0.03 | −0.23 |
| Density (2022) | −0.06 | −0.12 | −0.09 | −0.04 | −0.15 | −0.24 |
| Settlement types (2021) | 0.09 | 0.15 | 0.11 | 0.04 | 0.09 | −0.12 |
| Population size (2021) | 0.04 | 0.25 | 0.25 | 0.04 | 0.02 | −0.21 |
| Foreigners (2021) | −0.02 | −0.22 | 0.28 | 0.00 | 0.11 | −0.10 |
| % of 20–29-year-old inhabitants (2021) | 0.42 | −0.11 | 0.36 | −0.15 | −0.09 | −0.23 |
| % of 65+-year-old inhabitants (2021) | −0.15 | 0.06 | −0.23 | 0.40 | 0.01 | 0.46 |
| % of low-qualification inhabitants (2021) | 0.09 | 0.08 | −0.13 | 0.23 | −0.34 | 0.37 |
| % of higher education inhabitants (2021) | −0.04 | −0.05 | −0.14 | −0.11 | 0.25 | −0.20 |
| Social income (2022) | 0.66 | −0.27 | 0.33 | 0.14 | 0.18 | −0.05 |
| Crime rate (2022) | 0.16 | −0.20 | 0.36 | −0.09 | 0.50 | 0.12 |

**Table 4.** *Cont.*

| | 2022—30 Municipalities with Higher % of Votes | 2022—30 Municipalities with Lower % of Votes | 2024—30 Municipalities with Higher % of Votes | 2024—30 Municipalities with Lower % of Votes | 2022–2024—30 Municipalities with Higher % of Growth | 2022–2024—30 Municipalities with Lower % of Growth |
|---|---|---|---|---|---|---|
| Average earnings (2021) | 0.11 | −0.02 | −0.03 | −0.09 | 0.12 | −0.15 |
| Income inequality (2021) | 0.17 | −0.16 | 0.20 | −0.22 | 0.35 | −0.27 |
| Unemployment (2021) | 0.45 | 0.17 | 0.66 | 0.37 | 0.28 | 0.09 |
| Gini coefficient (2021) | 0.24 | −0.06 | 0.26 | −0.20 | 0.25 | −0.24 |
| Purchasing power (2021) | −0.04 | −0.08 | 0.12 | −0.10 | 0.28 | −0.25 |
| Public investment (2021) | 0.29 | −0.11 | 0.03 | −0.48 | −0.04 | −0.33 |
| Funding executed under Portugal 2020 (2022) | 0.28 | −0.15 | 0.51 | −0.17 | −0.19 | −0.14 |
| Funding executed per capita under Portugal 2020 (2022) | 0.10 | −0.19 | −0.13 | −0.63 | −0.02 | −0.62 |
| | Very low | Low | Moderate | High | | |

Thus, in 2022, we observe that in the thirty municipalities where the percentage of votes for the CHEGA party was highest, there is a strong correlation with the number of beneficiaries of social insertion income, with the vote growing in municipalities where the allocation of this type of subsidy has a higher proportion per thousand inhabitants of working age (Figures 8–10). Similarly, we observe that the ideas have greater adherence in places with higher unemployment and younger voters. Some similar factors, notably foreigners, also explain a lower percentage of votes in the thirty municipalities with less expressive voting for CHEGA.

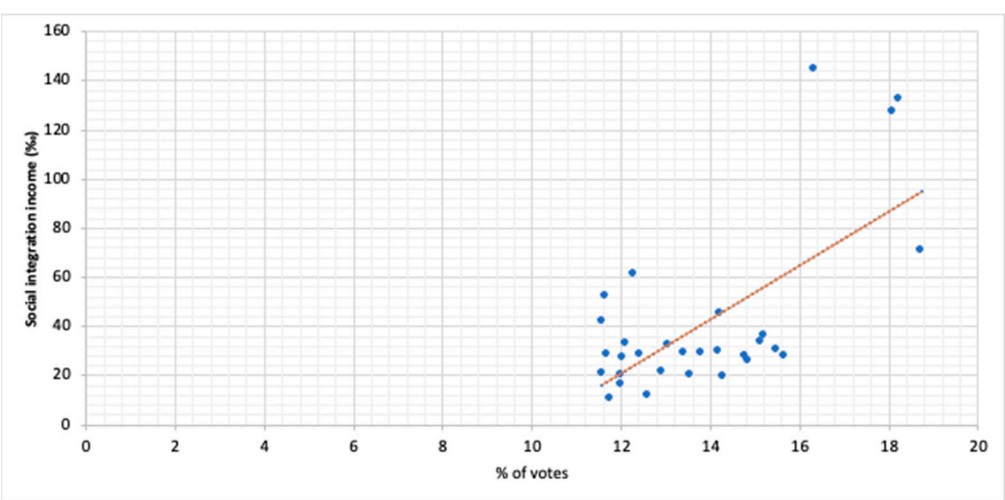

**Figure 8.** High correlation of votes (cluster-2022—30 municipalities with a higher % of votes) with social income beneficiaries. Source: own elaboration, using data from [34,35].

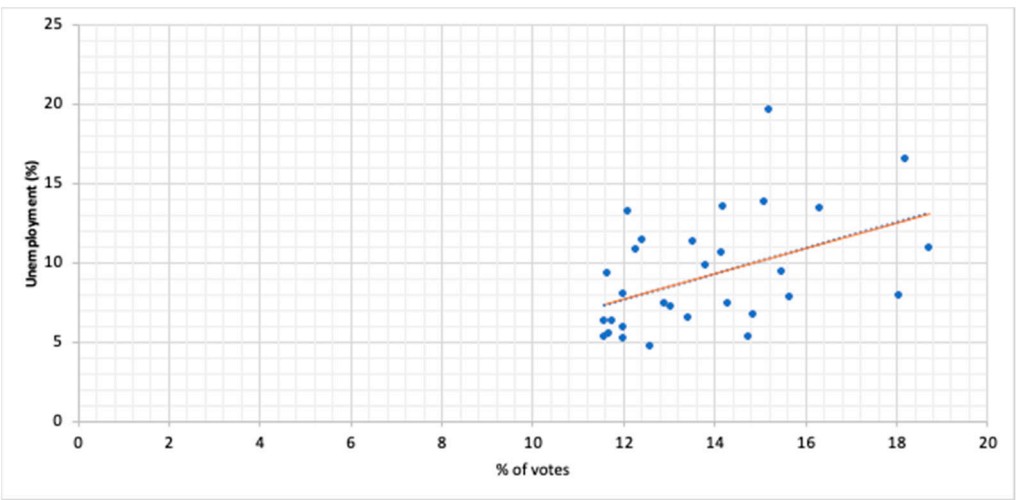

**Figure 9.** Moderate correlation of votes (cluster-2022—30 municipalities with higher % of votes) with unemployment. Source: own elaboration, using data from [34,35].

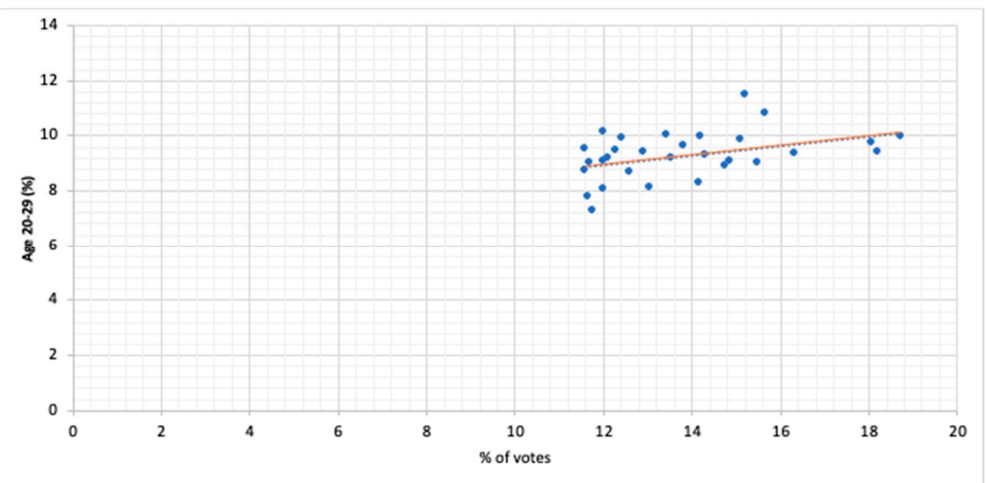

**Figure 10.** Moderate correlation of votes (cluster-2022—30 municipalities with higher % of votes) with younger population. Source: own elaboration, using data from [34,35].

In 2024, there are three factors with a high or moderate correlation in the set of thirty municipalities with the highest percentage of votes for CHEGA (Figures 11–13). The main one concerns the unemployment rate, with an increase in voting being observed in this context. The others point to a percentage increase in voting in municipalities with higher abstention rates and a higher volume of execution of community funds, associated with criticism of their effectiveness at solving some problems (particularly in the Algarve). In municipalities with lower voting, we verify that the voting increases within the elderly and less qualified, and it diminishes when public investment is higher (Figures 14 and 15).

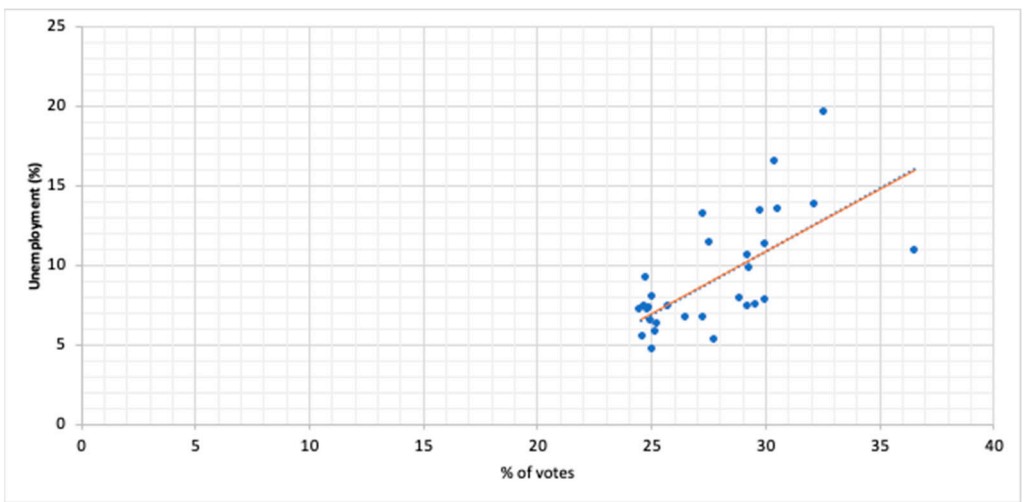

**Figure 11.** High correlation of votes (cluster-2024—30 municipalities with higher % of votes) with unemployment. Source: own elaboration, using data from [34,35].

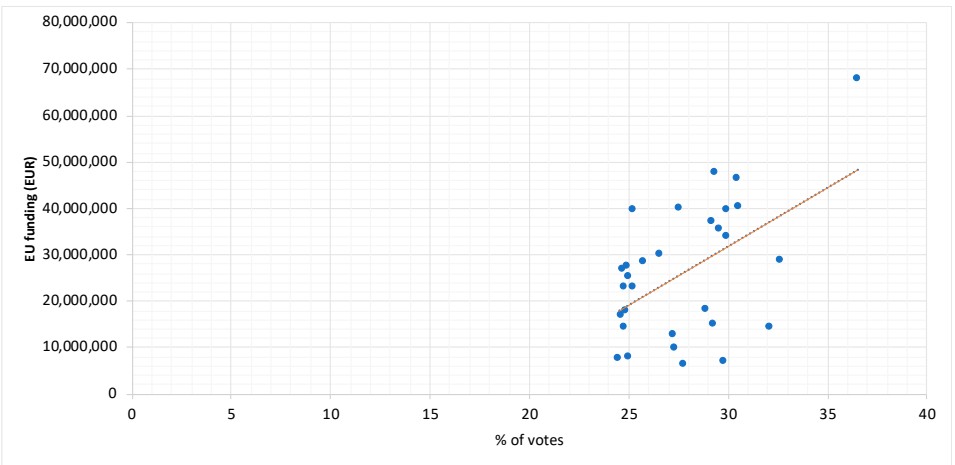

**Figure 12.** Moderate correlation of votes (cluster-2024—30 municipalities with higher % of votes) with EU funding (total amount). Source: own elaboration, using data from [34,35].

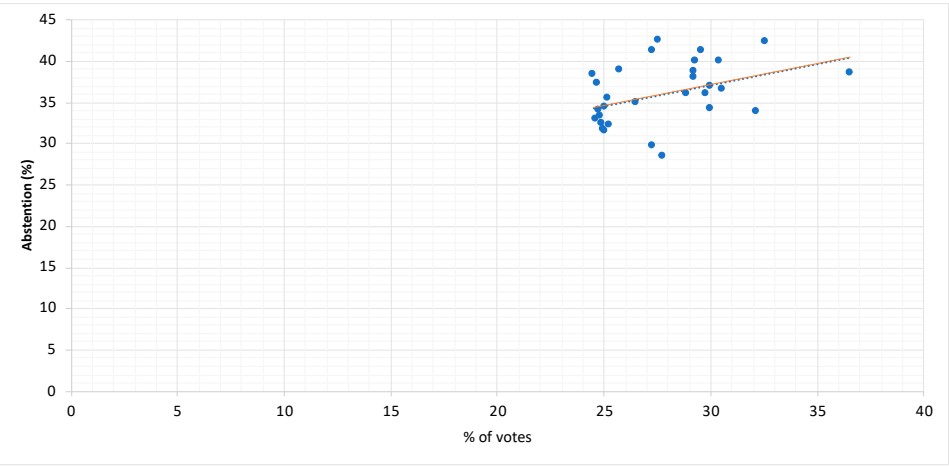

**Figure 13.** Moderate correlation of votes (cluster-2024—30 municipalities with higher % of votes) with the abstention rate. Source: own elaboration, using data from [34,35].

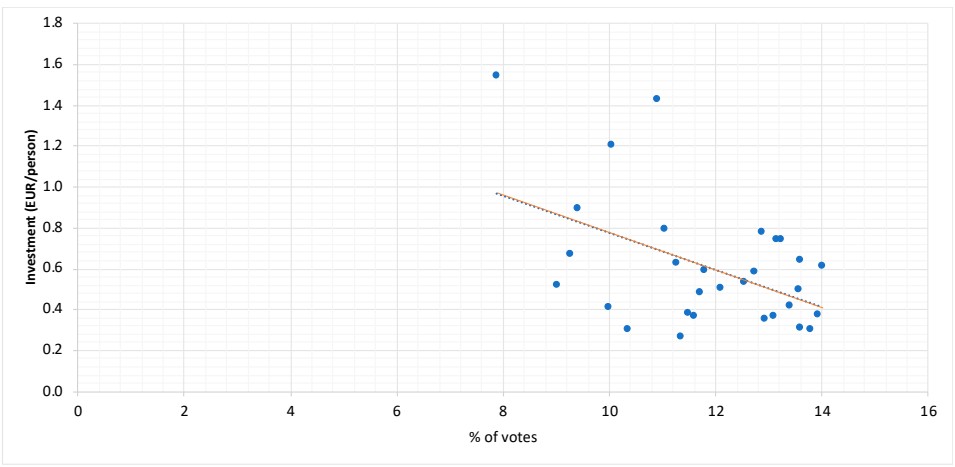

**Figure 14.** Moderate correlation of votes (cluster-2024—30 municipalities with lower % of votes) with public investment. Source: own elaboration, using data from [34,35].

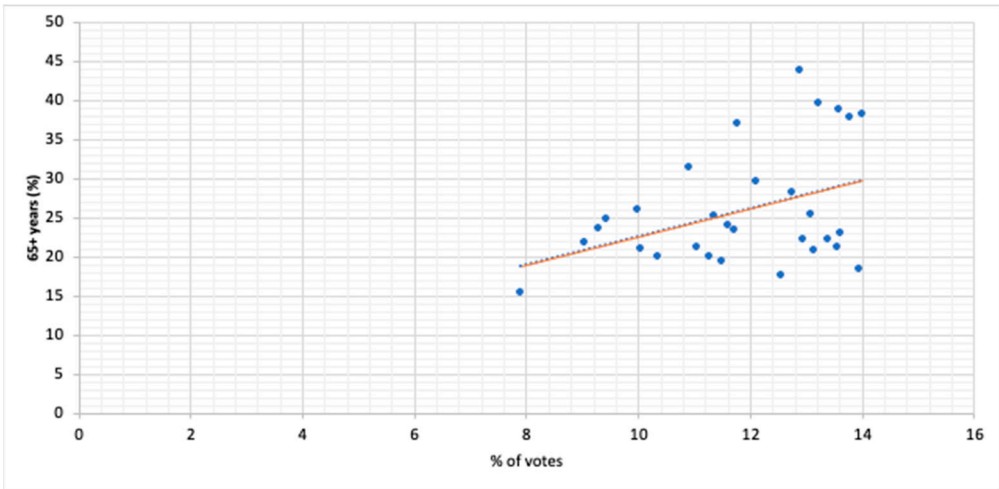

**Figure 15.** Moderate correlation of votes (cluster-2024—30 municipalities with lower % of votes) with elderly population. Source: own elaboration, using data from [34,35].

The analysis of the growth in voting between 2022 and 2024 shows a strong relationship with the crime rate (Figure 16), unemployment, and several economic and social issues in municipalities, with respect to the highest percentage of votes for the far right. Among the thirty municipalities with the smallest variation, we observe a strong relationship with the elderly (Figure 17) and less qualified.

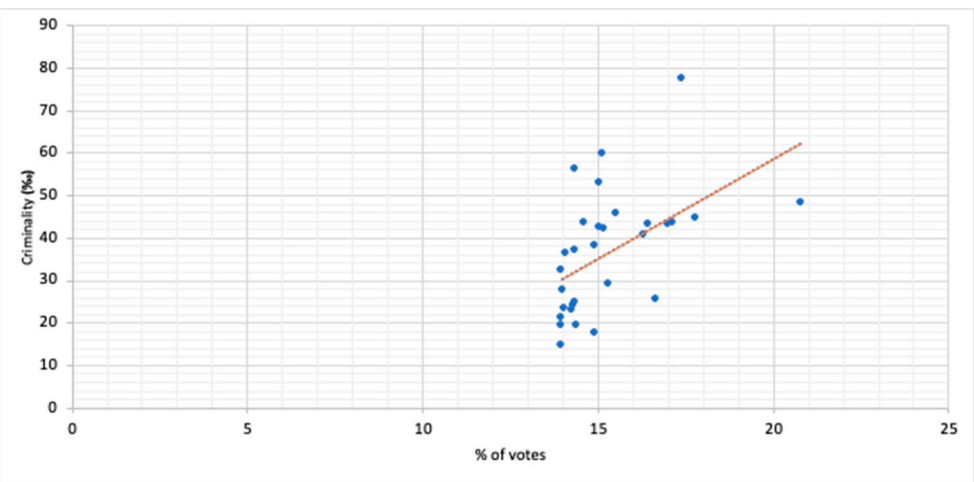

**Figure 16.** Moderate correlation of votes (cluster-2022–2024 growth—30 municipalities with lower % of votes) with crime rate. Source: own elaboration, using data from [34,35].

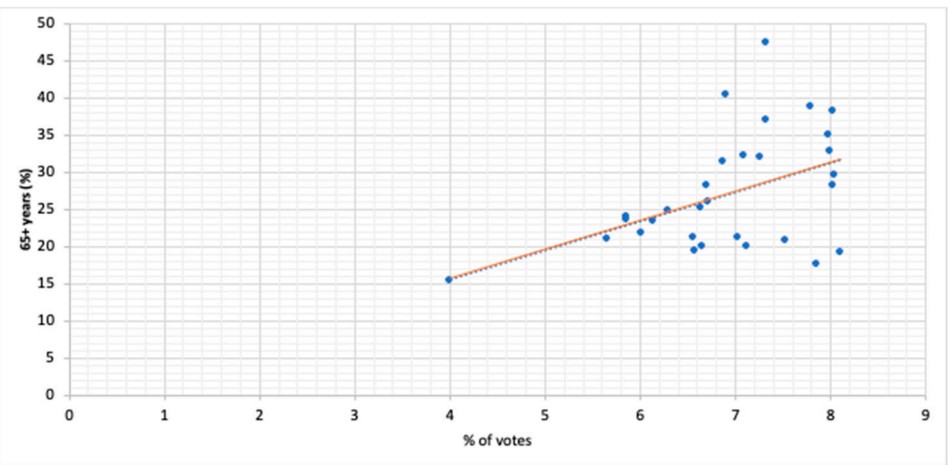

**Figure 17.** Moderate correlation of votes (cluster-2022–2024 growth—30 municipalities with lower % of votes) with elderly population. Source: own elaboration, using data from [34,35].

## 5. Discussion: Unraveling the Dynamics of CHEGA's Electoral Success in Portugal

The research findings not only shed light on the specific dynamics propelling the electoral success of Portugal's CHEGA party but also offer valuable insights into the broader context of right-wing populism. Through a synthesis of survey responses and statistical analyses, we navigate the intricate interplay of socio-economic, demographic, and territorial factors that contribute to the rise of this populist entity. In exploring these complexities, it becomes evident that CHEGA's trajectory is not a mere outlier but rather a reflection of a global trend in populist surges.

The survey uncovers a striking trend among CHEGA voters, indicating a significant endorsement of radical proposals and receptive stance towards the party's demagogic discourse. Anti-immigration and anti-welfare state policies find resonance among this electorate, highlighting a profound validation of populist narratives. This aligns with broader international patterns where right-wing populist movements often frame their platforms around anti-establishment sentiments, the rejection of liberal democratic norms, and appeals to the anxieties of the common populace.

Contrary to expectations, the statistical analysis reveals a nuanced correlation landscape between CHEGA voting patterns and various socio-economic indicators. While some connections exist, particularly in areas related to crime rates, foreign population percentages, and beneficiaries of social insertion income, these correlations are moderate

to weak. This departure from conventional expectations challenges prevailing narratives surrounding right-wing populist support, emphasizing the complexity and multifaceted nature of electoral dynamics.

Delving into the 2024 legislative elections, a notable shift in correlation strengths emerges. Areas experiencing higher unemployment rates become focal points for CHEGA's growth, challenging conventional wisdom and underscoring the significance of overlooked territories in the party's appeal. This shift echoes global trends where economic distress and discontent in regions facing neglect have become breeding grounds for populist sentiments. The issue of the foreign population remains relevant. In the sixteen municipalities with a proportion of foreign population exceeding 10%, CHEGA obtained an average vote of 23.6%.

Examining the broader determinants of populism, the Portuguese scenario does not always align with international examples. Conventional characteristics of populist supporters, such as older age, lower education levels, and economic disadvantage [10,18–20], do not find complete resonance in the CHEGA voter profile despite being relevant in the municipalities where CHEGA has less votes. These factors, often dubbed the "holy trinity" of populist voting [11,22], echo trends observed in other populist movements globally but present specificities in Portugal.

Intriguingly, this research illuminates a direct connection between public investment choices and the party's electoral success. Municipalities with lower per capita public investment under the Portugal 2020 framework witness an uptick in CHEGA support, and the votes are especially relevant within populations with higher education. This underscores the importance of understanding how political discontent becomes spatially expressed, with public investment strategies influencing voter sentiments.

The term "populism" remains a subject of intense debate within political science. CHEGA, like other right-wing populist parties globally, pitches "the people" against perceived elites, creating a dichotomy of "us" against "them". This antagonistic framing is a common thread running through populist movements, revealing a shared strategy to galvanize support by tapping into public frustrations.

Drawing parallels with international examples, the rise of CHEGA mirrors trends witnessed in countries like Hungary, Italy, the United Kingdom, and France. In these cases, right-wing populist movements capitalized on anti-immigrant sentiments, economic insecurities, and discontent with established political norms. The Brexit referendum in the UK, Fidesz in Hungary, and the National Rally in France exemplify how nationalist fervor and anti-establishment sentiments reshaped political landscapes.

The Portuguese analysis also brings attention to the evolving nature of populist support. Traditionally associated with older, less-educated, and economically disadvantaged individuals, populist support now transcends these demographics. The research hints at a shifting landscape, where discontent in neglected territories becomes a potent force in the populist surge.

In conclusion, the rise of CHEGA in Portugal encapsulates a complex interplay of socio-economic, demographic, and territorial factors, reflecting a broader international trend in right-wing populist movements. The nuanced correlation patterns, the shifting dynamics in the 2024 elections, and the alignment with global populist characteristics underscore the need for a multifaceted understanding of electoral dynamics. This research not only highlights the deciphering of specificities of Portuguese populism but also contributes to the broader conversation on the global rise of right-wing populist movements.

## 6. Conclusions: Navigating the Crossroads of Discontent and Development

The persistent challenges of poverty, economic decay, and limited opportunities plaguing declining and lagging-behind areas worldwide have become the crucible for a profound discontent that transcends social divides. These so-called "places that don't matter" [13] have witnessed a seismic shift in political expression, unexpectedly manifested through the ballot box in a wave of territorial-based political populism. This phenomenon challenges

existing wellbeing paradigms in both less dynamic and prosperous regions, prompting a critical reevaluation of development strategies.

Based on the information analyzed, the growth of votes for the CHEGA party can be attributed to several factors. Firstly, its ability to capture new voters, as evidenced by the significant percentage of respondents who had never voted for CHEGA before. Secondly, positive evaluations of the party's performance by its electorate may contribute to voter retention and growth. As for the territorially differentiated growth of this ideology, it appears to be associated with various economic and social variables. These include dissatisfaction with the governance of mainstream parties, as indicated by issues such as healthcare, education, and living conditions. Additionally, the party's focus on specific concerns, such as immigration and crime, may resonate more strongly in areas experiencing higher levels of unemployment and lower public investment, leading to increased support in these regions. Therefore, it is more and more clear that the "places that don't matter" are becoming more and more relevant in this context.

The research on Portugal's CHEGA party provides a lens through which we can discern broader global trends. The unexpected electoral success of populist movements, not solely rooted in socio-economic factors but deeply entwined with territorial dynamics, calls for urgent attention to the plight of neglected territories. Understanding the nuanced correlation patterns uncovered by the research becomes crucial for crafting effective, place-sensitive territorial development policies.

The call for action extends beyond conventional compensatory measures. It necessitates a paradigm shift towards policies that tap into the latent economic potential of these overlooked regions, fostering real opportunities and addressing the root causes of neglect and decline. The research suggests that a place-sensitive approach [2,12,29,30], steering clear of past development strategies focused on welfare, income support, and large-scale investments, is paramount for success.

The urgency of this intervention is underscored by the need to confront the long-term trajectories of low, no, or negative growth and to counteract the effects of industrial decline and brain drain. The research posits that place-sensitive policies can be the linchpin in addressing the present economic decline, bolstering human resources, and generating employment opportunities that form the bedrock of the geography of EU discontent.

As we contemplate the future, the imperative is clear: a shift towards place-sensitive policies is not just a development strategy but a safeguard for European integration and the stability that underpins the continent's historical period of relative peace and prosperity. The rising tide of anti-establishment voting threatens the very fabric of European unity, making it paramount to adopt measures that address the root causes of discontent and provide tangible solutions to those residing in the "places that don't matter".

In the landscape of future research, the exploration of place-sensitive policies demands greater attention. Understanding the intricacies of how these policies can effectively tap into the economic potential of neglected regions, provide genuine opportunities, and counteract the rise of anti-establishment voting is a frontier that requires comprehensive investigation. Additionally, comparative analyses with other global contexts experiencing similar populist surges can enrich our understanding and contribute to a nuanced, globally relevant framework for addressing the challenges posed by neglected territories.

In closing, the intersection of discontent and development requires bold, innovative, and place-sensitive solutions. The echoes of neglect not only reverberate within the borders of Portugal but resonate with the broader global narrative of political populism. It is through the use of thoughtful, nuanced policies that prioritize neglected regions, which provide pathways to economic revitalization and genuine opportunities, that we can navigate the crossroads of discontent and forge a more resilient, inclusive future for all.

**Funding:** This research was funded by Fundação para a Ciência e Tecnologia grant number UIBD/00736/2020.

**Institutional Review Board Statement:** Not applicable.

**Informed Consent Statement:** Not applicable.

**Data Availability Statement:** The data presented in this study are openly available in https://datarepositorium.uminho.pt/ (accessed on 27 April 2024).

**Conflicts of Interest:** The authors declare no conflicts of interest.

## Notes

1   Governmental monotony refers to a predominance of parties from the central government bloc, with a moderate orientation to the left (socialist party) or right (social democratic party), with a reduced expression (until recent years) of parties or more extremist movements.

2   The EU goal of Economic, Social, and Territorial Cohesion is defined in articles 174 to 178 of the *Treaty on the Functioning of the European Union*.

3   Articles 349th and 355th of the *Treaty on the Functioning of the European Union* (TFUE).

4   ENOUGH in English.

5   The total amount of votes in CHEGA in these five municipalities was 6762: 1063 in Serpa; 1001 in Moura; 1106 in Salvaterra de Magos; 400 in Vila Nova da Barquinha; and 3192 in Vila Verde.

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
