# Peer review of "Discontent, Populism, or the Revenge of the “Places That Don’t Matter”? Analysis of the Rise of the Far-Right in Portugal"

_societies, doi:10.3390/soc14060080_

Round 1

Reviewer 1 Report

Comments and Suggestions for Authors

This is a theoretically and empirically rich paper, and it makes a coherent and important argument. A few comments/questions

1. "n terms of the country's governance, these fifty years have been marked by great stability" - what does this stability look like? Is it not more likely that this stability may have come at the cost of the intra-country and regional inequalities this paper describes?

2. There are terms used such as “governmental monotony”, for which a definition is required.

3. "Despite this scenario, Portuguese society remains relatively inexpressive in the organization of formal or informal movements of public dissent. The exception has been the struggles of professional organizations (teachers, nurses, security forces, for example) or in housing matters." - Are there other explanations for the relative absence of collective organising and dissent?

4. Line 111 - "So, discontent is a potent force that can shape a nation's political destiny" - Is the author suggesting that this discontent is new? And if not, what is so particular about the new “wave” of discontent? And what are the discontents in areas that have materially more? How do intra-regional disparities play out beyond and besides populist discourse and politics?

5. "The survey data indicates that issues such as perceived corruption within the government, disillusionment with traditional political parties, and concerns about economic and social degradation strongly influence voter choices." - why has the political left been unable to talk on these issues, to provide a powerful alternative narrative?

Author Response

Dear Reviewer,

Thank you for carefully reading the text, comments and suggestions you made. They were all considered, introducing some additional notes into the text, with the following rationale.

This is a theoretically and empirically rich paper, and it makes a coherent and important argument. A few comments/questions

"In terms of the country's governance, these fifty years have been marked by great stability" - what does this stability look like? Is it not more likely that this stability may have come at the cost of the intra-country and regional inequalities this paper describes?

A sentence was added to reinforce that idea “This stability is associated, among other aspects, with the importance of metropolization and suburbanization processes, combined with a strong dependence on community funds and periods of reduced social contestation, despite the continuous worsening of internal territorial asymmetries.”

There are terms used such as “governmental monotony”, for which a definition is required.

A footnote was added “Governmental monotony refers to a predominance of parties from the central government bloc, with a moderate orientation to the left (socialist party) or to the right (social democratic party), with a reduced expression (until recent years) of parties or more extremist movements.”

"Despite this scenario, Portuguese society remains relatively inexpressive in the organization of formal or informal movements of public dissent. The exception has been the struggles of professional organizations (teachers, nurses, security forces, for example) or in housing matters." - Are there other explanations for the relative absence of collective organising and dissent?

An explanation text was added: “In Portugal, the relatively low frequency of public protests can be attributed to several factors. Historically, the country experienced a long period of authoritarian rule under the Estado Novo regime, which suppressed dissent and left a legacy of cautiousness towards public demonstrations. Additionally, the strong influence of the Roman Catholic Church and a culture that values social harmony over conflict can contribute to a preference for resolving issues through dialogue rather than protest. Economic factors also play a role, as the focus on stability and recovery after the financial crises has led many to prioritize economic security over public activism. However, exceptions do exist, particularly among professional groups.”

Line 111 - "So, discontent is a potent force that can shape a nation's political destiny" - Is the author suggesting that this discontent is new? And if not, what is so particular about the new “wave” of discontent? And what are the discontents in areas that have materially more? How do intra-regional disparities play out beyond and besides populist discourse and politics?

Further explanation was added “Discontent is becoming more and more relevant in Portugal, due to recent social and economic issues and some generation changes. In fact, economic issues such as austerity measures, rising inequality, and housing shortages have led to increased public discontent. These pressing issues have mobilized people to take to the streets and demand change, highlighting the growing role of public dissent in shaping political outcomes. Younger generations, who did not experience the dictatorship, are more inclined to express their dissatisfaction and demand political changes. They are also more connected through social media, which helps organize and amplify protests.”

"The survey data indicates that issues such as perceived corruption within the government, disillusionment with traditional political parties, and concerns about economic and social degradation strongly influence voter choices." - why has the political left been unable to talk on these issues, to provide a powerful alternative narrative?
The political left has been unable to talk on these issues, and to provide a powerful al-ternative narrative, as they have supported the government between 2015 and 2019 and some of their policies after that.

Reviewer 2 Report

Comments and Suggestions for Authors

Dear author(s)!

The article deals with an important topic and offers a good insight into the complexity of processes that influence the rise of far-right and populist parties. The connection with geographical context is interesting and worth examining.

However, I believe that the article, in its present form, has some issues that need to be addressed.

The main question posed by the author is why the increase of votes for the CHEGA party happened between 2022 and 2024 and what are the economic and social variables explaining the territorial differences in the growth of this ideology. While the empirical part offers an insight into the territorial differences and contributes to research on why people decide to vote for far-right populist parties, the question of why Portugal witnessed such a drastic change in two years remains unanswered. The reasons for these questions to remain unanswered might also be in the selection of methodology and data.

The first part of the empirical evidence focuses on the survey among CHEGA voters. The use of this survey and its characteristics would need to be better explained. At this moment, it is not clear:

-        How reliable or representative is the sample? Here, a description of the population would be needed. There is doubt on the quality of the sample, since the author mentions on many occasions that older and less educated are more likely to vote for far-right populist parties but the sample taken in the survey mostly has younger people.

-        Why is the survey for 2022 relevant for addressing the research questions of the paper? While the data on reasons for voting for CHEGA can offer an important insight into the topic, I do not see the added value for the article in other parts of the survey (figures 3 and 4).

The second part of empirical evidence focuses on finding statistical correlations between different municipality indicators and votes. This analysis sheds light on the important geographical indicators that contribute to the support for far-right and populist parties. However, it is less clear how the findings contribute to explaining the rise of support in Portugal. Here I point out two things:

-        The indicators include data from 2021 or 2022. It would be worth including newer data to see if changes in some indicators contributed to the rise of votes.

-        Some of the correlations seem to be influenced by outliers (for example figures 8, 12, 14). This should be addressed and explained better. Furthermore, the information about the statistical significance of the correlations is missing.  

Furthermore, some factual mistakes need to be rechecked and corrected:

-        On page 2, the authors note that the CHEGA party achieved 48 elected deputies while the result at the moment is 50 deputies (https://www.parlamento.pt/sites/EN/Parliament/Paginas/Election-results.aspx).

-        In the methodological chapter on page 6, it is written that the survey ‘took place between September 1 and October 30 2024’ which is not possible.

-        In the same paragraph on page 6 the dates do not match – on one occasion, it is written elections of 2022, and on the other, the elections of 2021.

I would also suggest some other additions or corrections to improve the clarity of the text:

-        More precision in terminology – the terms far-right and populist parties seem to be used as a synonym. Are they really the same? In this context, the theoretical background can be improved and more clearly specified (for example: what role do traditional parties and institutions and their insufficient responses to some of the crises play in this?).

-        A few sentences on the establishment of CHEGA and its relationship with previous far-right parties would be useful for clarification in the introduction chapter.

-        When placing the case of Portugal within the wider European perspective, more contemporary examples can be mentioned (f.e. the Netherlands or Sweden) or the examples could be better connected with the case of Portugal.

-        I would suggest more elaboration on the existing explanations of the rise in votes for CHEGA – so far other explanations are only quickly mentioned as post-election media analyses on page 8.

Author Response

Dear Reviewer,

Thank you for carefully reading the text, comments and suggestions you made. They were all considered, introducing some additional notes into the text, with the following rationale.

Dear author(s)!

The article deals with an important topic and offers a good insight into the complexity of processes that influence the rise of far-right and populist parties. The connection with geographical context is interesting and worth examining.

However, I believe that the article, in its present form, has some issues that need to be addressed.

The main question posed by the author is why the increase of votes for the CHEGA party happened between 2022 and 2024 and what are the economic and social variables explaining the territorial differences in the growth of this ideology. While the empirical part offers an insight into the territorial differences and contributes to research on why people decide to vote for far-right populist parties, the question of why Portugal witnessed such a drastic change in two years remains unanswered. The reasons for these questions to remain unanswered might also be in the selection of methodology and data.

The first part of the empirical evidence focuses on the survey among CHEGA voters. The use of this survey and its characteristics would need to be better explained. At this moment, it is not clear:

-        How reliable or representative is the sample? Here, a description of the population would be needed. There is doubt on the quality of the sample, since the author mentions on many occasions that older and less educated are more likely to vote for far-right populist parties but the sample taken in the survey mostly has younger people.

The text includes data on the profile of survey respondents, who end up representing all ages. It intended to analyze the perceptions, evaluations and voting intentions of the electorate in CHEGA. The questionnaire was made available freely, the response was not limited to a specific profile (demographic, social or economic), and only responses from those who confirmed having voted for CHEGA in 2022 were validated.

-        Why is the survey for 2022 relevant for addressing the research questions of the paper? While the data on reasons for voting for CHEGA can offer an important insight into the topic, I do not see the added value for the article in other parts of the survey (figures 3 and 4).

The survey of 2023 (relating to voters in 2022) is essential to understand the nuclear profile of CHEGA voting. Figure 3 adds on the territorial dimension, and figure 4 the scope of CHEGA party, much more on a national scale rather than on a proximity basis. A small comment was added to the text.

The second part of empirical evidence focuses on finding statistical correlations between different municipality indicators and votes. This analysis sheds light on the important geographical indicators that contribute to the support for far-right and populist parties. However, it is less clear how the findings contribute to explaining the rise of support in Portugal. Here I point out two things:

-        The indicators include data from 2021 or 2022. It would be worth including newer data to see if changes in some indicators contributed to the rise of votes.

Most of the indicators are produced within the Census, every 10 years (2001, 2011, 2021…) We used the most recent indicators.

-        Some of the correlations seem to be influenced by outliers (for example figures 8, 12, 14). This should be addressed and explained better. Furthermore, the information about the statistical significance of the correlations is missing. 

The statistical significance is now explained (The Pearson correlation coefficient measures the linear relationship between two variables. To determine the statistical significance of this correlation, we need to test the hypothesis that there is no correlation between the variables (0). The Pearson correlation coefficient provides a measure of the strength and direction of the linear relationship between two variables. Values close to ±1 indicate strong relationships, while values close to 0 indicate weak or no linear relationship. The value of 1 indicates a perfect positive linear relationship between the two variables. As one variable increases, the other variable increases in exact proportion. The value of -1 indicates a perfect negative linear relationship. As one variable increases, the other variable decreases in exact proportion.).

We conducted a Grubb's test to identify potential outliers in our dataset, which comprised information from 308 municipalities. The purpose of the test was to determine if any data points significantly deviated from the others, potentially affecting our analysis. The results of the Grubb's test indicated that the presence of outliers was minimal, suggesting a high level of consistency and reliability in our data. Consequently, the test confirmed that the influence of outliers on our overall findings is negligible, allowing us to proceed with confidence in the robustness of our dataset.

Furthermore, some factual mistakes need to be rechecked and corrected:

-        On page 2, the authors note that the CHEGA party achieved 48 elected deputies while the result at the moment is 50 deputies (https://www.parlamento.pt/sites/EN/Parliament/Paginas/Election-results.aspx).

We had a reference to four deputies elected from internacional circles, which was not concluded when the paper was submitted. Information was changed.

-        In the methodological chapter on page 6, it is written that the survey ‘took place between September 1 and October 30 2024’ which is not possible.

It was a typo. Corrected to 2023

-        In the same paragraph on page 6 the dates do not match – on one occasion, it is written elections of 2022, and on the other, the elections of 2021.

It was a typo. Corrected to 2022

I would also suggest some other additions or corrections to improve the clarity of the text:

-        More precision in terminology – the terms far-right and populist parties seem to be used as a synonym. Are they really the same? In this context, the theoretical background can be improved and more clearly specified (for example: what role do traditional parties and institutions and their insufficient responses to some of the crises play in this?).

Some clarification was added “Far-right parties typically advocate for conservative or reactionary positions, em-phasizing nationalism, anti-immigration policies, and traditional values while often displaying authoritarian tendencies and prioritizing national sovereignty. On the other hand, populist parties appeal to the frustrations of ordinary people against elites, using anti-establishment rhetoric and claiming to represent the "true" voice of the people. While some far-right parties may incorporate populist strategies to broaden their appeal, pop-ulism itself can manifest across the political spectrum and isn't inherently tied to specific ideological positions like those of far-right parties (Guiso et al., 2017, Hagemeister, 2022). In Portugal, far-right parties totally incorporate populist strategies and discourses.

Traditional parties and institutions play a significant role in shaping the political landscape and public opinion. When these entities fail to adequately address societal crises or effectively respond to the concerns of the population, it can create a vacuum that populist and far-right parties often exploit (Bursztyn et al., 2020). Insufficient responses from traditional parties and institutions may include perceived or actual neglect of issues such as economic inequality, immigration, cultural identity, and political corruption. When people feel marginalized or disenfranchised by the mainstream political estab-lishment, they may turn to populist or far-right alternatives that promise to address their grievances and provide simplistic solutions (Valentim, 2021). Additionally, a lack of trust in established institutions, coupled with a sense of frustration with the status quo, can further fuel the appeal of populist and far-right movements. These movements often capitalize on this discontent by presenting themselves as outsiders who will challenge the existing power structures and restore the voice of the "forgotten" or "ignored" segments of society.”

-        A few sentences on the establishment of CHEGA and its relationship with previous far-right parties would be useful for clarification in the introduction chapter.

It was added on the introduction “CHEGA, a far-right party in Portugal, shares similarities with previous far-right parties in its emphasis on nationalist rhetoric, anti-immigration policies, and criticism of mainstream political elites. Like other far-right movements, CHEGA has capitalized on populist sentiments, portraying itself as a voice for the "ordinary people" against perceived threats to national identity and sovereignty. While CHEGA's specific platform and tactics may differ from previous far-right parties, its emergence reflects broader trends of populist and nationalist sentiment gaining traction across Europe.”

-        When placing the case of Portugal within the wider European perspective, more contemporary examples can be mentioned (f.e. the Netherlands or Sweden) or the examples could be better connected with the case of Portugal.

We have added some explanation on it “The rise of CHEGA in Portugal shares some similarities with the political landscape in these countries. It also connects strongly with some contemporary developments like the Netherlands and Sweden, where far-right parties have also experienced varying degrees of growth. In the Netherlands, the Party for Freedom, led by Geert Wilders, has gained prominence with its anti-Islam and anti-immigration stance, appealing to voters disillusioned with mainstream politics. Similarly, in Sweden, the Sweden Democrats have capitalized on concerns about immigration and multiculturalism to become a significant political force. These parties, like CHEGA, have tapped into populist sentiments and criticized traditional political elites for failing to address the anxieties of the population. While the specific contexts and issues may differ across these countries, the broader trend of far-right parties gaining ground by exploiting societal tensions and grievances against established institutions is evident.”

-        I would suggest more elaboration on the existing explanations of the rise in votes for CHEGA – so far other explanations are only quickly mentioned as post-election media analyses on page 8.

We have elaborated on it, adding some ideas of public and mediatic discussion in Portugal. The results also refer to it. “The rise in votes for CHEGA, can be attributed to a combination of socio-economic factors, political disillusionment, and cultural anxieties, considering the main rationale of public discussion within the media and academic meetings. Firstly, economic uncertainty and inequality have left many Portuguese citizens feeling marginalized and overlooked by the mainstream political establishment. CHEGA's nationalist rhetoric and promises to prioritize the interests of ordinary Portuguese people resonate with those who perceive themselves as being left behind by globalization and economic reforms. Secondly, there's a growing disillusionment with traditional political parties, seen as out of touch with the concerns of the average citizen and mired in corruption scandals. CHEGA's leader, André Ventura, has capitalized on this sentiment by positioning himself as an outsider willing to challenge the status quo and shake up the political establishment. Additionally, cultural anxieties about immigration and national identity have fueled support for CHEGA, as the party advocates for stricter immigration policies and portrays itself as a defender of Portuguese heritage and values. By tapping into these societal tensions and offering simplistic solutions to complex issues, CHEGA has managed to attract a significant share of the electorate, particularly among those disenchanted with mainstream politics.”

Round 2

Reviewer 2 Report

Comments and Suggestions for Authors

Dear author(s)!

Thank you for including the suggested improvements in the paper,  I think the quality has improved. I only have a few minor details I would like to point out before publication:

  • You corrected the information on the number of elected CHEGA deputies on page 9, but not on page 2.
  • The added paragraphs are very informative; however, I would suggest adding some references and sources to support your statements and strengthen your arguments (for example, on pages 2 and 5, references from media articles, party programs or statements would be useful).
  • In chapter 3 (methods), the additional comment on the survey is good; however, I would suggest adding a footnote with information about the population (the total number of people that voted for the CHEGA party in the 2022 elections in the municipalities where the survey took place).
  • The p-value is added for Grubb’s test; I would advise adding it also for tables 3 and 4.

Author Response

Dear author(s)!

Thank you for including the suggested improvements in the paper,  I think the quality has improved. I only have a few minor details I would like to point out before publication:

Dear Reviewer. Thanks for you comments. All suggestions were considered.

  • You corrected the information on the number of elected CHEGA deputies on page 9, but not on page 2.

It was corrected.

  • The added paragraphs are very informative; however, I would suggest adding some references and sources to support your statements and strengthen your arguments (for example, on pages 2 and 5, references from media articles, party programs or statements would be useful).

References were introduced.

  • In chapter 3 (methods), the additional comment on the survey is good; however, I would suggest adding a footnote with information about the population (the total number of people that voted for the CHEGA party in the 2022 elections in the municipalities where the survey took place).

Footnote was added.

  • The p-value is added for Grubb’s test; I would advise adding it also for tables 3 and 4.

Thanks for the suggestion. It was added to the tables.